# Research on sublevel open stoping recovery processes of inclined medium-thick orebody on the basis of physical simulation experiments

Jin Wu [1,2]*

**1** Key Laboratory of Ministry of Education on Safe Mining of Deep Metal Mines, Northeastern University, Shenyang, China, **2** Business Administration Department, Liaoning Radio and TV University, Shenyang, China

* 77823457@qq.com

**Citation:** Wu J (2020) Research on sublevel open stoping recovery processes of inclined medium-thick orebody on the basis of physical simulation experiments. PLoS ONE 15(5): e0232640. https://doi.org/10.1371/journal.pone.0232640

**Data Availability Statement:** All relevant data are within the manuscript and its Supporting Information files.

**Funding:** This study was funded by the National Science Foundation of China (51874068) and the

## Abstract

The gently inclined medium-thick orebody is generally viewed as difficult to extract. This paper presents a sublevel open stoping using long-hole with back filling method, particularly for inclined medium-thick orebodies. This method not only changes the temporal sequence of ore stoping and realizes transportation using gravity and trackless devices, but also improves production capacity and efficiency. Nonetheless, this method also has some disadvantages; for example, the orebody has more contact with country rock, and the method has a more complex loss and dilution process. This paper starts with the analysis of the results of physical simulation experiments that examine the interactive relationship among the lower stope footwall dip angle, the footwall surface roughness, draw point interval and production blast ring and concludes their influence on the ore-recovery ratio in each production cycle. Secondly, based on physical simulation results, the multivariate linear fit was carried out on the SPSS using the dimensional analysis method. Thus a statistical model was developed for investigating the influence of gently inclined medium thick orebody, the footwall dip angle, the footwall roughness and interval of draw points on the recovery ratio, which can accurately forecast the ore-recovery ratio under different parameters in the physical simulation process. The optimal structural parameters obtained from physical simulation and statistical analysis was then applied to industrial experiments. Based on the 3D laser scanning during in-suit experiments, it was found that the lower stope had an 82% recovery ratio and an 18% dilution ratio, while the upper stope had a recovery ratio of 85% and a dilution ratio of 12%. Moreover, the production capacity could be up to 600t/d. The physical simulation and industrial experiments both demonstrate that the new mining method can be adopted for the safe and efficient mining of gently inclined medium-thick orebodies.

Fundamental Research Funds for the Central
Universities (N160107001, N180701016).

**Competing interests:** The authors have declared
that no competing interests exist.

## Introduction

Orebody with a dip angle of 20–50˚ and a vertical thickness of 5–15 m is called inclined
medium-thick orebody [1]. Inclined medium-thick orebody accounts for 5–9% of non-ferrous
ore reserves, 18% of iron ore reserves, 30% of reserves in gold mines, and over 70% of reserves
in phosphate rock [2]. Since the dip angle of the footwall is very gentle, the blasted muck can-
not be removed by gravity. Besides, LHD cannot enter the field of operations due to the limita-
tion posed by the ramp angle. As a result, the falling ore is usually transported with other
methods such as scraper transportation or blast-throwing haulage. Unfortunately, the scraper-
transport method is not highly mechanized or automated, so it has small production capacity,
requires more operators, and is complex to organize. Explosive throw movement is also prob-
lematic as it is poor in accuracy and difficult to control the throwing distance.

After mining has been completed, a large area of the hanging wall, or top roof of the
medium-thick orebody is left exposed; this leads to a heavy supporting workload, which conse-
quently diminishes the safety of the job-site. Thus, inclined medium-thick orebody is known
as being typically difficult to mine.

According to domestic and foreign statistics on inclined medium-thick orebody, room-
and-pillar mining method takes up 55% of the mines, bottom-hopper caving method occupies
18% of the cases, cut-and-fill method accounts for 24% of mines, and explosive throw move-
ment accounts for about 3% [3]. The room-and-pillar method is used most frequently for ore-
bodies that are inclined 20–30˚ [3–5], but the applied technology will vary depending on the
ore body's thickness. For orebodies with less than 5m in thickness, single-layer recovery is usu-
ally used, while those exceeding 5m are usually extracted using multi-layer recovery. In addi-
tion, if the roof is not stable enough in multi-layer mining, the bench method can be used for
top-down mining. In this method, the hanging wall surface is exposed after the roof or hanging
wall has been reinforced using appropriate roof support methods, and then short-hole mining
equipment is used for mine recovery. Two examples of this approach in practice can be seen at
Heqing Manganese Mine and HouZhuang Gold Mine in China. If the hanging wall is firm
enough, the inverted bench method can be adopted to extract ore in a bottom-up level
sequence, where the bottom drift is first crosscut for partial ore removal, and the remaining
ore is used as a layered platform for upper layer mining operations.

The short-hole shrinkage or sublevel chamber-mining methods are usually adopted for
mining orebodies with a 30–50˚ incline and less than 10 m in vertical thickness [6–8]. Both
methods require an auxiliary ore-drawing structure, such as footwall funnel, to recover the ore
remnants from the footwall. The case of this method may be found in Nanshan mining area in
Sanshan county, Xinjiang Uyghur Autonomous region, the No. 2 ore body of Anqing Copper
mine. When the orebody is more than 10m in vertical thickness [9–11], the sublevel caving
method is most often used. This method also requires an auxiliary recovery structure to
recover ore remnants from the footwall and bottom floor, as can be seen at the most mines in
China and the Chambishi Copper Mine in Zambia. The backfilling method can be used for
deposits with high mining values and whose hanging walls and entire orebodies are not stable.
Examples of this technique in practice include Xincheng Gold Mine and Jiaojia Gold Mine in
China.

The above mining methods all have some shortcomings in their implementation. For exam-
ple, in the production process, when the room-and-pillar method or the large mechanized
equipment cannot be employed, it makes the process difficult to manage, and results in a small
production capacity and low operating efficiency. In addition, workers are forced to work
under the exposed stope roof, which can potentially be unsafe.

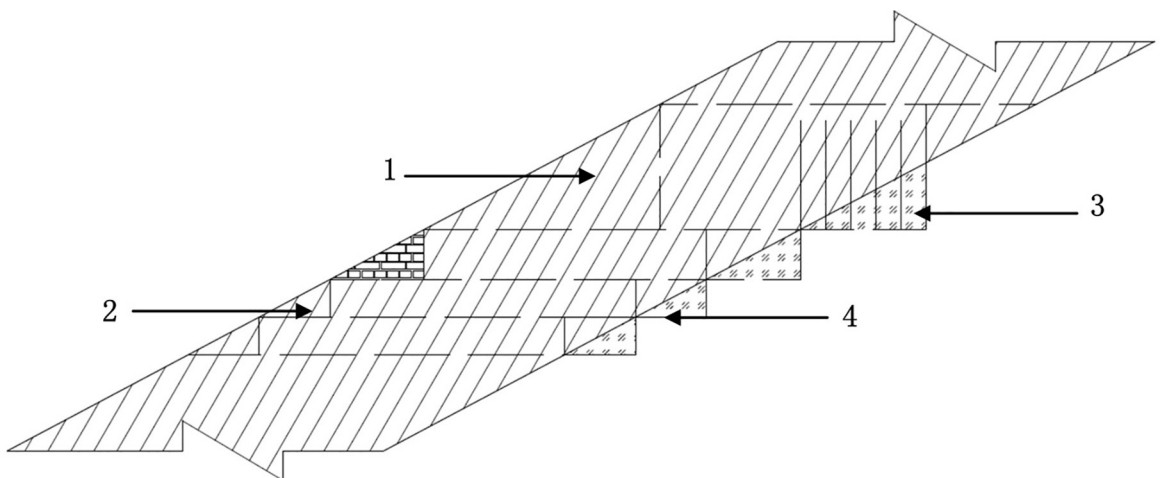

**Fig 1. Schematic of inclined ore body mining loss and dilution position.** 1 Triangle region of hanging wall in long-hole mining; 2 Triangle region of hanging wall in short-hole mining; 3 Footwall regions in long-hole mining; 4 Footwall regions in short-hole mining.

Both the shrinkage method with a footwall hopper ore-drawing structure and the sublevel-stopping mining method have high mining-cut workloads and ratios (up to 30–50 m/kt). Furthermore, a large number of pillars are left over once mining has been completed, which can result in an ore-loss ratio as high as 20–30% [9]. The improved sublevel caving method leaves two triangular areas between the hanging wall and the footwall-ore junction (see Area 1 & Area 3 in Fig 1), which results in an ore-recovery ratio of only 60–70% and a dilution ratio of up to 30–40%. In addition to a diminished ore-recovery ratio and a high dilution ratio, recovery projects in the surrounding rocks also require a very large mining workload. As the orebody tilts to the contact surface between the orebody and the rock body at the hanging wall, a triangular area will similarly occur if the short-hole stoping method is used, as is shown in Areas 2 & 4 in Fig 1. During the recovery process, the waste rock mixed ratio can be up to 50%; however, if recovery is not conducted, the amount of ore loss can account for 15–20% of the total ore deposit. In summary, all the conventional methods for mining inclined medium-thick ore bodies have significant disadvantages, including high loss and dilution ratios, small-scale production capacity, a low level of mechanization, and potentially unsafe working conditions, among others. All of these drawbacks make it difficult to achieve large-scale caving and are not efficient or safe.

### Recovery rate of inclined medium-thick ore body in sublevel open stoping using long-hole with back filling mining method

Considering the above problems associated with inclined medium-thick orebody stoping, a sublevel open stoping using long-hole with back filling mining method arranged along the ore strike is proposed. The stope, which are approximately 30 m in length, are arranged vertically and are spaced 10m from one another, with each sublevel using a bottom-up mining sequence. The ore body in each sublevel is extracted with a two-step approach: first, the lower stope is exploited, and then the upper stope is mined after the lower stope has been backfilled. During the exploitation of lower stope, sublevel transport drifts are constructed along the orebody. The sublevel height is 10 m and production drilling drift are constructed at the ore body and surrounding rock junctions. Loading crosscuts are then constructed every 6–8 m to connect the drilling drift. During extraction, the slot is constructed, and upward-directed fan drillings

are made in the drilling drift, which is intended to conduct the retreating blast from the central slot to the two sides of the stope.

The extraction process is completed by LHDs, which are used to remove the ore from the loading crosscut drifts. After the lower chamber stope is extracted, backfilling materials with a cement-sand ratio of 1:4 are dropped into the goaf from a drift in the upper sublevel. The extraction of the upper stope begins when the filling strength is reached. Rock drifts are then constructed and access drifts are extended to form an ore-removal structure that uses the trackless equipment. Similar drilling and blasting steps are repeated to extract the ore in the upper chamber stope in the same level. Compared to the drift and fill mining method, this operational program has the following merits: (1) by changing the form of mining, blasted ore can outflow on their own via gravity, which solves the problem of ore haulage in inclined ore bodies; (2) the stope is divided into two parts—during the extraction process in the lower frame stope, the surrounding rock in the upper frame stope is not exposed, which allows extraction to realize under the protection of the upper frame stope and avoids the premature collapse and depletion of the upper frame stope; (3) it solves the loss and dilution problems inherent to the mining of triangular ore on the upper stope of inclined ore bodies using the drift and fill method; (4) long-hole blasting and mechanized transportation by trackless equipment greatly enhance efficiency and productivity, reduce the number of working stopes, and simplify the operation organization; (5) All the drilling and blasting operations are conducted in a stable tunnel rather than under a large exposure area, which improves the work safety.

After blasting, LHDs are used to remove the ore via the loading drifts. Since not all of the buckets can enter the loading drift, a triangular muck pile, known as 'end residue', is left untransported at the tip of the ore vein (as shown in Fig 2A). Ridge remnants can be found between loading drifts (shown in Fig 2B), and the greater their distance from the space of loading drifts, the greater amount of the remnants. If the angle of the ore body at the footwall is less than the ore-rest angle, it will decrease the flow of the ore body; the smaller the angle, the poorer the granular flow, and the greater amount of remnant left at the footwall (Remnants as shown in Fig 2C). In order to recover the blasted muck remaining on the footwall, part of the surrounding rock at the footwall is blasted and then blasted into the mucks as shown in Area D in Fig 2. The mixture of the waste rock increases the dilution rate. The increment of the dilution rate is determined by the quantity of mixed waste rock and hence is related to the footwall side blast hole angle and lower plate flatness.

Ore loss and dilution are used to characterize the ore-reducing quantity and ore grade-descent level in the mining process, both of which are important indexes for evaluating the advantages and disadvantages of a mining method. Numerical simulation is an important means of studying ore loss and dilution [12, 13], and it is a useful tool for revealing the micro-mechanical mechanism in the ore dilution and loss process. However, the numerical simulation process is greatly limited by the difference between calculated parameters and boundary conditions within the in-situ field. Physical simulation experiments [14–21] are also an essential method of studying the loss and dilution process in production. Experiments based on scaled simulation are conducted to simulate the geometry and mechanical processes of production, from which ore-flow patterns can be obtained for statistical indicators of ore loss and dilution. In addition, 3D laser scanning systems [22–24] can duely provide the volume and shape of the stoped area, and have become an effective contactless means to assess the actual industrial indicators.

Therefore, based on physical simulation, this paper investigates how ore recovery and dilution ratio are affected by the footwall dip angle, footwall surface roughness, production drift interval, and ore-drawing burden. Based on dimensional analysis, a calculation model is built to assess the loss of inclined medium-thick orebody during long-hole caving. This calculation

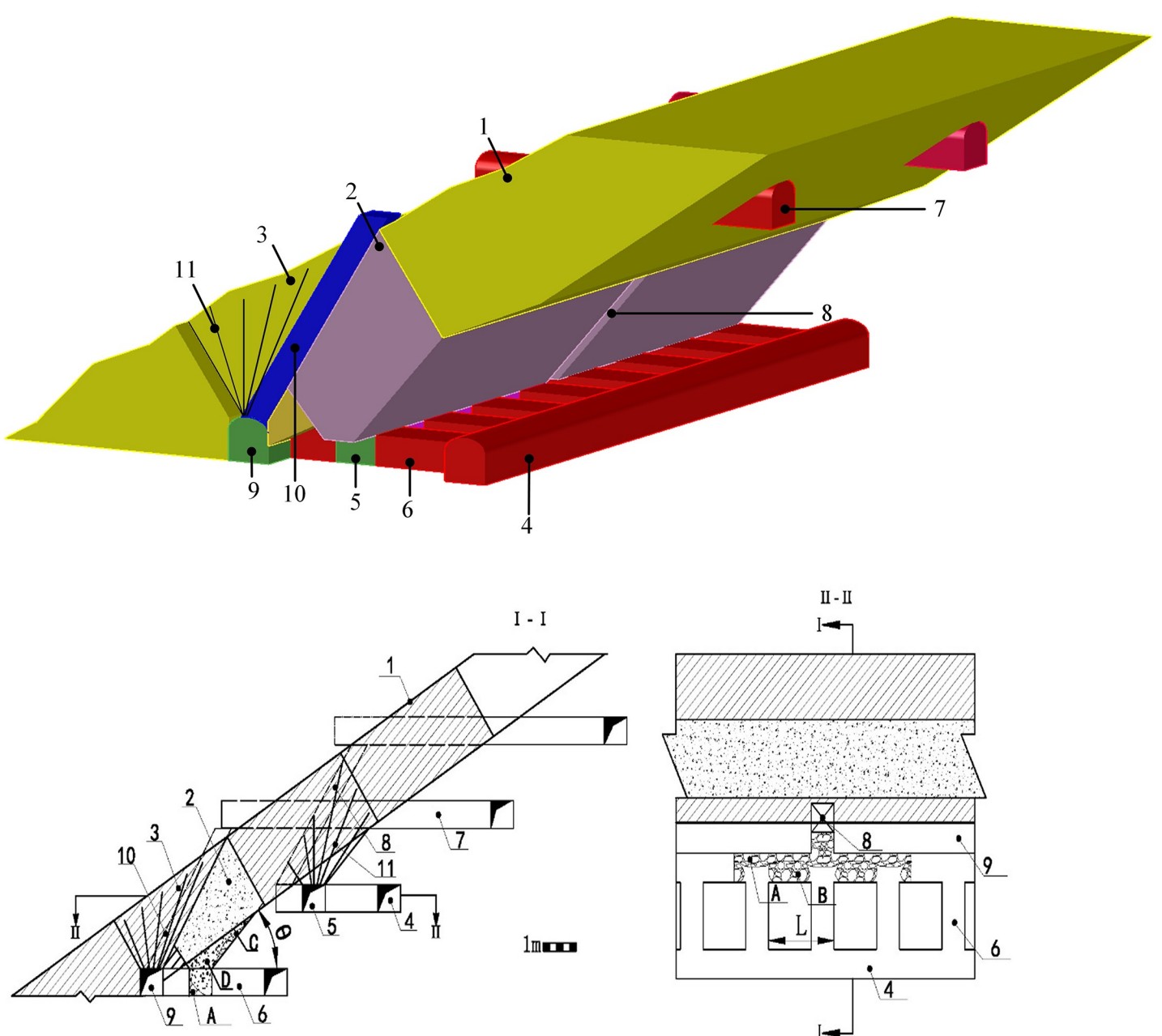

**Fig 2. The schematic of sublevel open stoping method with double-drilling roadway.** (A) section sketch of DSLOS mining method; (B) 3D model of the DSLSOS mining method 1 Parallelogram stope; 2 Lower stope; 3 Upper stope; 4 Sublevel transport drift; 5 Drilling drift of lower stope; 6 Loading crosscut; 7 Sublevel filling drift; 8 Cutting raise of lower stope; 9 Drilling drift of upper stope; 10 Cutting raise of upper stope; 11 Fan drilling; A: drift end losses; B: Ridge remnants; C: Sidewall remnants at footwall; D: Mixed waste rock at footwall.

model provides the basis for the optimal structure parameters applicable for industrial experiments and the specific measures to reduce ore loss and dilution presented in this paper. The experimental results of this study were comparatively analyzed with the industrial experimental results using a CMS. It was done in order to verify the adaptability of the physical simulation and the computational model that provided the theoretical support for the viability of mining inclined medium-thick orebodies using the segmented-framing long-hole stoping method.

## Materials and experiments

### Determination of similar conditions

Physical simulation tests need to meet the following assumptions [25, 26]: (a) gravity flow only involves research on non-adhesive, crushing-shaped blocks, and the loose ore rocks can move slowly under gravity; (b) loose ore rocks are heterogeneous and isotropic; (c) special weak boundaries are not considered in the ore-drawing process; and (d) the secondary crushing of ore and rock blocks are not considered. Based on the above assumptions, the similar conditions of physical simulation in this paper are set as follows: (1) test models are geometrically similar vis-a-vis dimensions (length, width, height), structure (the number of drawing points and size, etc.), and granular particle components; (2) the model loading density is equal to the prototype's loose bulk density, $\lambda_p = \lambda_g = 1$; (3) the proportion of time is related to the proportion of length scales, $\lambda_t = \lambda_l^{1/2}$; (4) the stress ratio is related to length, $\lambda_o = \lambda_r = \lambda_l = 1$; (5) residual friction angles are equal, $\lambda_{or} = 1$; (6) the internal friction angle is equal to the friction angle of the loose rock, $O_w = O$.

First, once blasting had been completed, an investigation of the in-site grain-size distribution of the ore was conducted. As shown in Table 1, 95% of the ore particle size was less than 0.8 m after the blast. By the 1:100 similarity ratios, the granular particles were too fine to be conducive to physical simulation experiment. However, with the 1:50 similarity ratio, 80% of the particles being 1.6 cm would be appropriate for physical simulation experiments. The original rock material is collected from the field and then crushed into grains in the laboratory. These grains are separated into different kinds based on their particle sizes as listed in Table 1. The rock sample is created by grains with different sizes according to the ratio in Table 1. Second, the bulk density of the test site after blasting was measured, and the backfilling density was ensured to be consistent with it during the simulation process. Thirdly, a physical ore-drawing model geometrically similar in size and structure to the designed mining scheme was developed using the designed similarity ratio. Finally, a scale-sized ore-drawing shovel, which corresponded to the ore bucket as per the similarity ratio, was designed in order to simulate the mining operations of a scraper. The above preparations were undertaken to ensure that the physical simulation tests were as similar as possible to the field production operation. The parameters of the experiments model are list in Table 2.

### Experiment scheme

According to the analysis of the mining method design in Section 2, lower stope footwall dip angle, at the point of blasting, the footwall flatness, production drift spacing, and ring burden after blasting all have a significant impact on ore loss and dilution. Therefore, a physical simulation test was designed in order to study the use of the sublevel open stoping using long hole with back filling mining method and the relationship between ore loss / dilution and the above factors. The simulation results of each factor is shown in Table 3. All factors were fully combined to design a total of 162 experimental programs; each experiment was repeated twice, and their average was taken as the final result.

**Table 1. Laboratory simulation of ore grain size composition.**

| In-situ Size/cm | <10 | 10~20 | 20~30 | 30~40 | 40~50 | 50~60 | 60~70 | 70~80 | 80~90 | 90~100 |
|---|---|---|---|---|---|---|---|---|---|---|
| Test Size/mm | <2 | 2–4 | 4–6 | 6–8 | 8–10 | 10–12 | 12–14 | 14–16 | 16~18 | 18~20 |
| Ratio | 0.73 | 10.19 | 10.85 | 11.07 | 15.76 | 16.20 | 16.81 | 14.42 | 3.81 | 0.15 |

**Table 2. Similarity factors of experiment model.**

| Dimension(m) | Scaled ratio | Parameters of model |
|---|---|---|
| Length(m) | 1:50 | 60 |
| Width(m) | 1:50 | 60 |
| Height (m) | 1:50 | 40 |
| Mining scheme (hours) | 1:7 | 2.5 |
| Bulk density (kg/m$^3$) | 1:1 | $1.5 \times 10^3$ |
| Ore-drawing shove (cm$^3$) | 1:50 | 60 |

## Physical simulation model

The physical simulation model, (shown in Fig 3), was composed of the floor, the lower stope, and the upper stope. The lower stope floor was a movable plate with an adjustable range of 30° to 55° to simulate the change of the lower footwall blast hole angle during extraction. Adjustable access drifts were built into the footwall plate along the strike of the orebody to simulate different spacing intervals, and a plate separator was set up at the footwall plate to simulate different ore-drawing burdens. The model was built to be 40cm high, 60cm wide, and 60cm long in order to simulate a stope measuring 30m in length, 30m in horizontal thickness, and 20m in height. Production drilling drift and loading drift were both rectangular in cross-section, and measured 7cm in width and 6.6cm in height to simulate an in-site field rock drift, 3.5m in width and 3.3m in height. The rock drift and access drift were made of iron sheets, while the remaining parts of the simulation model were made of transparent acrylic plates as it provided easier external observation of the ore's granular movement during the mining process. Each component of the model was connected and fixed by screws, pins, and grooves.

## Experimental process

In light of the experimental scheme, the ore extraction experimental model was assembled, and the burden plate and production drifts were moved into place. Once the model had been assembled, the simulated ore was weighed and evenly loaded into the burdens. During the experiment, ore-burden plates were drawn in sequence to simulate blasting in accordance with the step set up in the simulation program. In the access drift, small rakes with a grip length of 400 mm and 50 mm in width were inserted 3 cm into the rock tunnel, and the ore body was extracted evenly at full-face along the loading drift. Such haulage was not terminated until they could no longer extract ore through the loading drifts. During the experiment, fine sandpaper, coarse sandpaper, and mineral ore were pasted to the footwall surface to simulate ore recovery ratios under different smoothness conditions.

## Result analysis of physical simulation experiment

### Influence of footwall side blasthole angle on recovery

As shown in Fig 4, when the footwall blast hole angle increased, the ore-loading quantity increased to about 40° before decreasing afterwards. The weight of the ore remnants gradually

**Table 3. Experimental program design.**

| Index | Experimental program | |
|---|---|---|
| 1 | Footwall side hole caving angle (°) | θ = 30/35/40/45/50/55 |
| 2 | Footwall friction coefficient (-) | f = 0.4/ 0.7/ 0.9 |
| 3 | Access drift interval (m) | d = 5/6/7.5 |
| 4 | Ore drawing burden (m) | w = 1/2/4 |

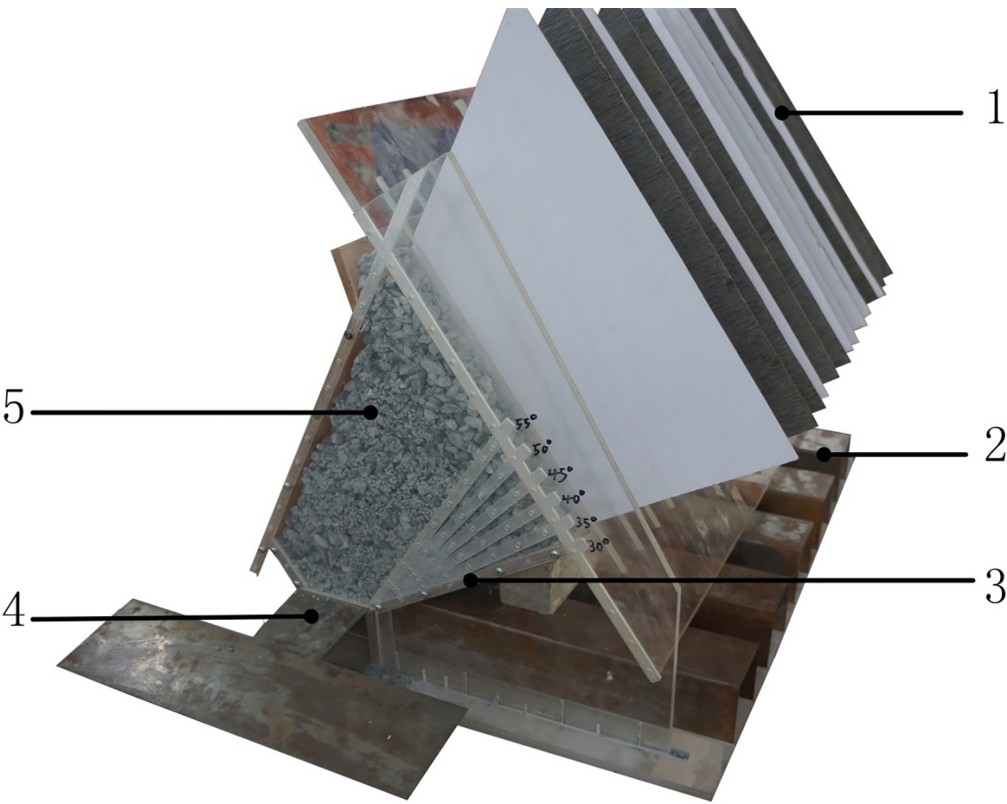

**Fig 3. Physical simulation model and its size.** 1 Burden plate; 2 Loading crosscut drift; 3 Adjustable footwall plate; 4 Drilling drift 5 Lower stoping.

decreased with the increase of the footwall side-hole angle and finally became stable at about 45˚ with a recovery ratio of 82%. Ore recovery increased as the footwall side-hole angle increased, reaching its maximum at 40˚; above 40˚, however, the recovery ratio decreased as the angle of the footwall side blast hole became larger, eventually stabilizing at above 45˚. Also, when the footwall side blast hole angle was less than 40˚, there was a linear relationship between the ore-recovery ratio and the side-hole angle. This is because, when the side-hole angle was small, the width of the model increased, which lead to a larger ore-loading quantity. This relationship is consistent with the phenomenon that in the production process the amount of blasted muck from one blast increases along with the increase of side blasted hole angle. During the ore-drawing process, the enlargement of footwall side blast hole angle enhanced blasted dispersion flow; therefore, the ore-recovery weight increased and the ore-remnant volume decreased. However, when the footwall blast hole angle was close to the repose angle of caving dispersion, it led to an ore flow of stable balance state with little change in the ore-recovery ratio. Thus, optimal ore recovery occurs when the angle of the footwall blast hole is 40˚.

## Effect of the footwall surface flatness

The ore recovery results after blasting under both smooth and rough footwall conditions are shown in Fig 5. Given the same footwall side blast hole angle, there was less ore-remnant volume with the smooth footwall surface than that with rough surface. When the side blast hole angle exceeded the natural ore-repose angle, there was no big difference in the number of ore

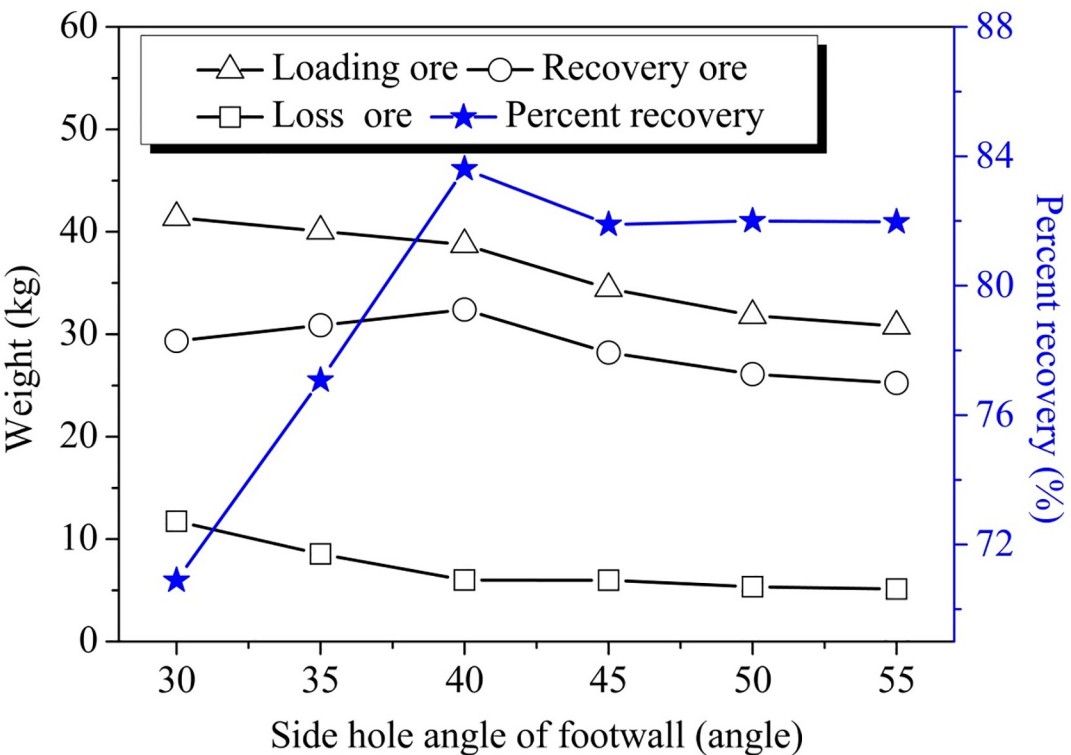

**Fig 4. Single step ore recovery.**

remnants at the footwall. In the ore-drawing process, when the footwall was smooth, the ore recovery ratio was 8–10% greater than that under rough conditions; however, regardless of the rough condition, the maximum value reached 40˚. Thus, after blasting, the smoothness of the footwall can greatly affect the ore-recovery ratio; the smoother the footwall-surface after blasting, the smaller the friction coefficient, and the better the ore-recovery effect.

## Influence of production ring burden

The ore-drawing results under three different ring burdens of 1 m, 2 m, and 4 m are shown in Fig 6. The results show that, given the same side blast hole angle, ore recovery under 1 m and 2 m burdens was typically 3–5% higher than that recovered under 4 m burden. Production ring burden mainly determines the quantity of blasted ore and the number of times that ore can be removed after each blast. Wider ring burdens result in a smaller window ore-drawing, which consequently means that there is less frequency to recover the ore; thus, causing the possibility for ore-recover diminished, and, by extension, the ore-recovery is relatively reduced. Meanwhile, the results of the physical experiments demonstrate that smaller ring burdens can produce greater recovery ratios.

## Influence of loading drift interval

In view of the LHD's inside radius and ore stability, the relationship between loading crosscut-spacing and ore-recovery was simulated under vein-spacing conditions of 5 m, 6 m and 7.5 m (results are shown in Fig 7). The same footwall side blast hole angle was used for all conditions, and the results indicated that the ore-recovery volume with the 5 m interval was larger than that with the 6 m and 7.5 m intervals; meanwhile, the remnant weight at the 5 m interval was

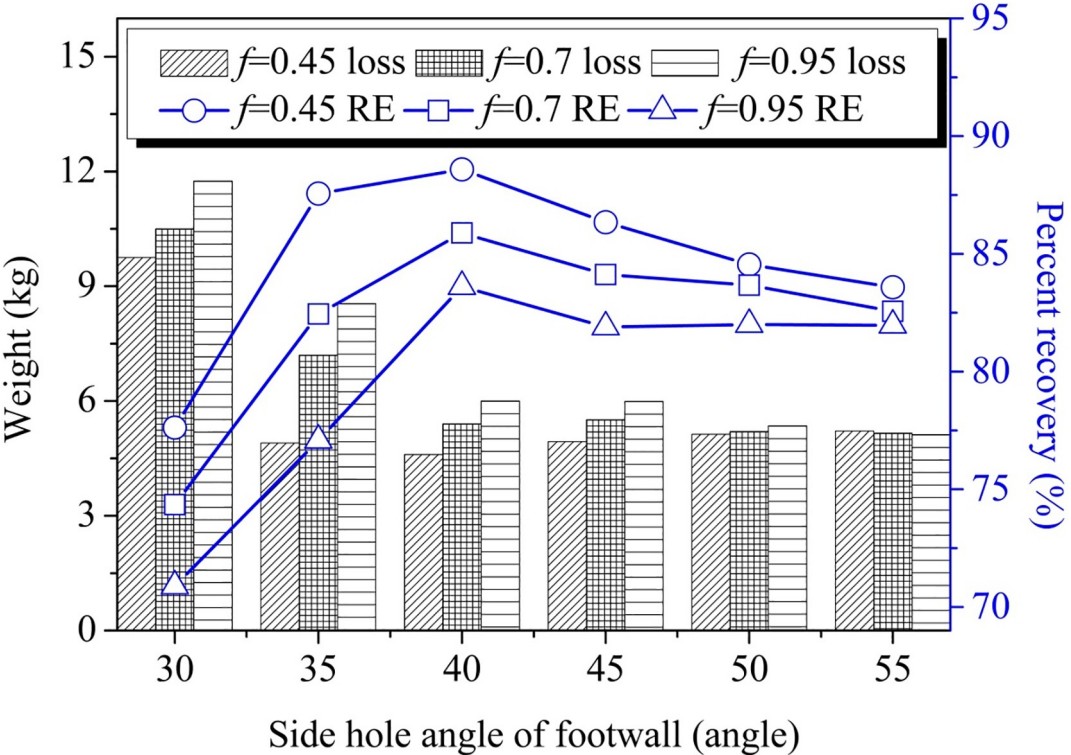

**Fig 5. Recovery variation along with footwall smoothness.**

smaller than that with the 6 m and 7.5 m intervals. The ore-recovery ratio with the 5m interval was about 5% higher than that with the 6 m interval. To sustain the stability of the pillar between loading drifts, the spacing between adjacent loading crosscuts should have a minimum width of 5 m in filed applications.

## Dimensional analysis

Dimensional analysis is a common method for establishing a mathematical model that describes the quantitative relationship between independent variables, and it has been widely used in the study of mining engineering [27–29] or example, in the mine-blasting operation-process, among many other related factors, Dehghani and Khandelwal [30, 31] used dimensional analysis to conduct a quantitative analysis of peak-particle vibration velocity and vibration frequency in the blasting seismic wave propagation process. Their development of a prediction model could provide guidance for production practice. In the study of mine gob caving-height, Zhao [32] (Kang, Kui, and Liang 2015) used dimensional analysis to establish a quantitative characterization describing the relationship between the gob caving height and many factors such as mining method, ore body thickness, mining depth, tectonic stress, and goaf volume. Zhao's quantitative characterization has subsequently been applied successfully to mining production. In the study of rock deformation and the fracture process, Fakhimi [33] used Multi-Dimensional Analysis to illustrate the relationship between the rock damage index and the initial stress, sample height, average diameter, and friction coefficient.

In physical process studies, measures that characterize physical properties are called dimensions. Some of the physical dimensions, which are called basic dimensions, include such things as length ($L$), time ($T$), force ($F$), and quality ($M$). Other physical quantities can be acquired by

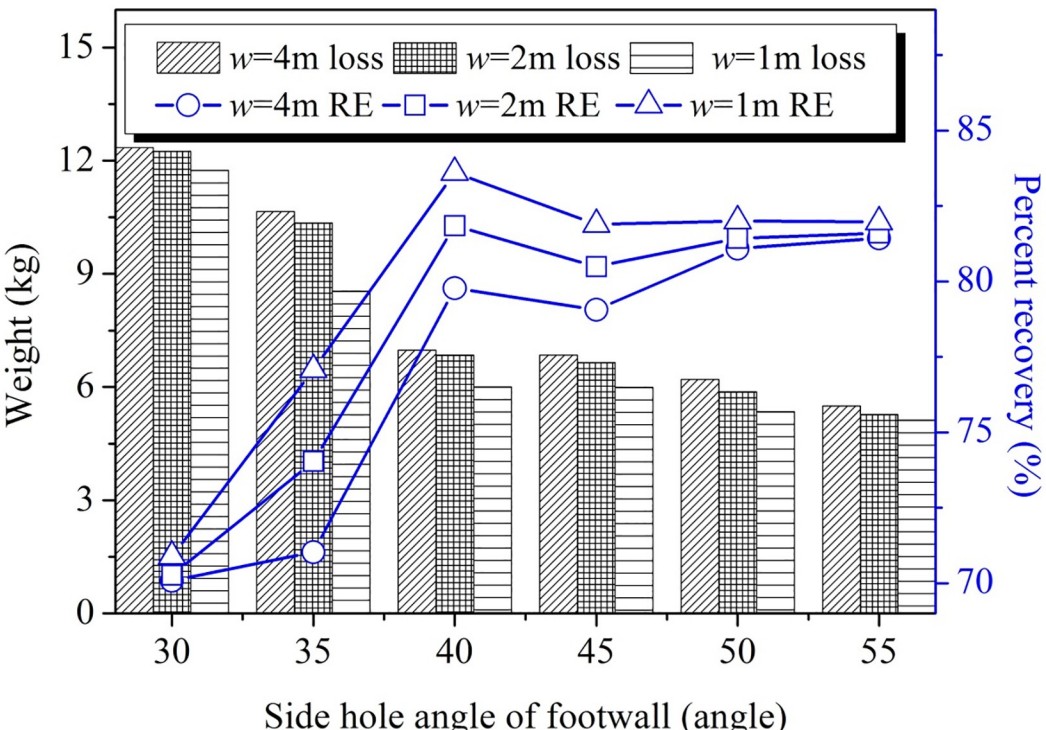

**Fig 6. Footwall side blast hole angle of single step chamber.**

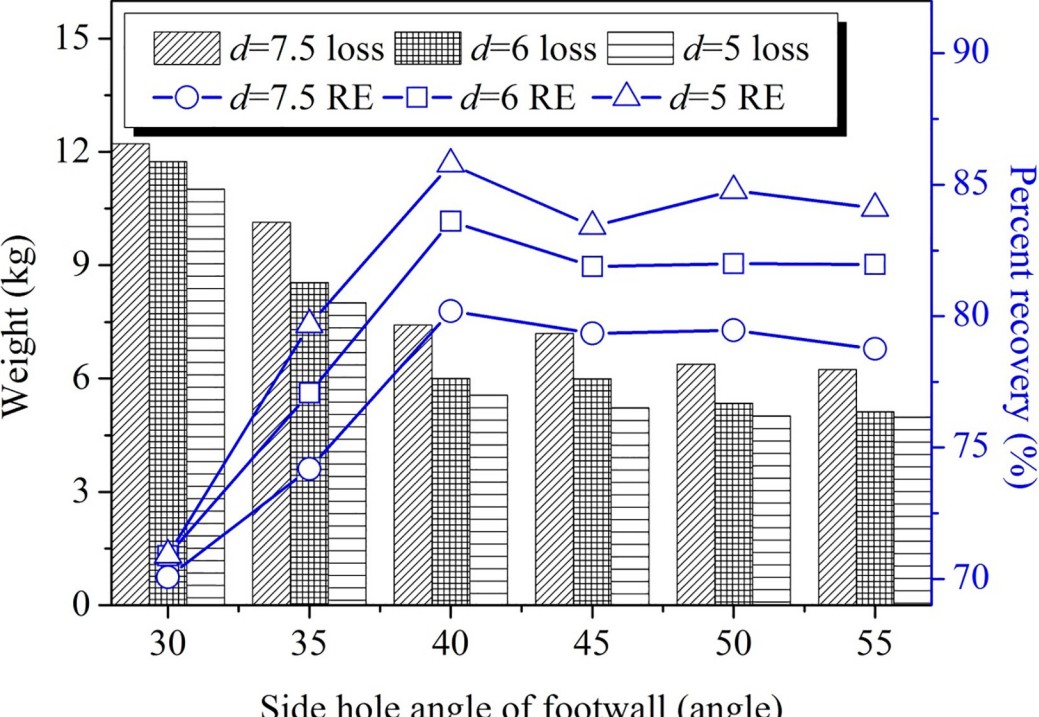

**Fig 7. Influence of different loading drifts space on ore drawing.**

determining involution product of basic dimensions. The dimensional analysis aims to remove the units of dimensional variables in the problem. Dimensional analysis is conducted in order to break down the variable dimensions relating to the problem into basic dimension systems, which are characterized by power, quality, length, or time; the purpose of dimensional analysis is to eliminate the unit in the problems under study; the process of dimensional analysis is first to select one or a few variables bearing units as the basic variables, then to use these basic variables to measure other variables bearing units, and ultimately to determine the relationship between various physical quantities by the volley principle of dimension in the laws of physics [29]. Based on the physical simulation experiments for the slowly inclined medium-thick orebody, there is a corresponding relationship between the ore-recovery ratio ($H$), the footwall blast-angle ($\theta$), footwall flatness ($f$), production drift spacing ($d$), ore-drawing burden ($w$), and ore-block length ($L$), as shown in Eq (1).

$$H = f(\theta, f, d, w, L) \quad or \quad f(H, \theta, f, d, w, L) = 0 \tag{1}$$

In the physical simulation experimental process, there are five factors that influence ore recovery, and these are listed in Table 4. Their dimensional expressions are: footwall blast-angle $[\theta] = 1$, dimensionless; footwall flatness unit $[f] = 1$, dimensionless; production drift-spacing unit $[d] = L$, dimension is length L; ore-drawing burden unit $[w] = L$, dimension is length L; ore block-length unit $[L] = L$, dimension is length L. Based on the above analysis, a dimensional matrix was built, which is shown in Table 5.

According to Table 5, the dimension matrix is expressed as follows:

$$K = \Delta = |1| = 1 \tag{2}$$

The value of the matrix determinant is 1, which shows that the five variables have a nonlinear correlation; the dimension matrix has an order 1. Therefore, according to the $\pi$ theory, the number (m) that influences the dimensionless quantity existing in factors of n = 5 is:

$$m = n - k = 5 - 1 = 4;$$

Therefore, Eq (1) can be expressed as follows:

$$\pi_1 = \theta;$$

$$\pi_2 = f;$$

$$\pi_3 = [L]^{\alpha}[L];$$

$$\pi_4 = [L]^{\beta}[L].$$

According to dimensional harmony theory, the sum of all dimensions is 0; therefore, the following equation can be acquired as:

$$\alpha + 1 = 0$$

$$\beta + 1 \equiv 0$$

Available: $\alpha = -1$, $\beta = -1$.

**Table 4. Dimension analysis parameters of ore simulation data set.**

| Number. | Parameter | Symbol | unity | Dimension |
|---|---|---|---|---|
| 1 | Footwall blast angle | θ | ° | - |
| 2 | Footwall flatness | *f* | - | - |
| 3 | Production drift spacing | d | m | L |
| 4 | Ore drawing burden | w | m | L |
| 5 | ore block length | L | m | L |

As a result, the five factors influencing recovery can be combined as the following four dimensionless dimension groups:

$$\pi_1 = \alpha;$$

$$\pi_2 = f;$$

$$\pi_3 = L/d;$$

$$\pi_4 = L/w.$$

From the above four dimensionless dimension groups, $\pi_3 = L/d$—which is the stope length divided by the production drift spacing—represents an explicit physical meaning of the number of ore-drawing locations. $\pi_4 = L/w$—which is stope length divided by the ring burden, and represents particle flow in production. Therefore, the influencing factors for ore-recovery ratio $H$ can be converted into: footwall dip of side blasthole angle $\theta$, footwall flatness $f$, the number of drawing points ($L/d$), and the number of blasting times ($L/w$). Figs 4–8 demonstrate that, during the ore drawing process, the ore recovery ratio shows an obvious linear relationship with the footwall side blasthole dip, footwall flatness, ore-route spacing, and the number of blasts. In the previous experiments, a total of 162 groups of experimental data were collected, out of which 95% were used in multivariate linear fitting, while 5% were used for the verification of fitting results. Multiple linear fitting [34] was carried out using an IBM SPSS 23, producing Eq (3):

$$H = 66.562 + 0.267\theta - 13.553f + 2.098(L/d) + 0.235(L/w) \tag{3}$$

The determination coefficient $R^2$ of the multiple linear-fitting results was 53.3%; after adjustment, it changed to 52%. Both were greater than50%, which indicates that there were few un-interpreted variables in the fitting results and that these results could cover most of the experimental data [1]. Variance analysis was then carried out on the fitting results; in the significance test of regression equation, the significance was 0, 0.05 less than the significance level. Thus, when the coefficients are not 0 at the same time in the regression equation, all the linear relationships between the explained variables and the explanatory variables are

**Table 5. Dimension matrix predicted by recovery.**

| Parameter | Recovery | Footwall blast angle | Footwall Flatness | Production Drift Spacing | Ore Drawing burden | Ore Block Length |
|---|---|---|---|---|---|---|
| F | 0 | 0 | 0 | 0 | 0 | 0 |
| L | 0 | 0 | 0 | 1 | 1 | 1 |
| 0 | 0 | 0 | 0 | 0 | 0 | 0 |

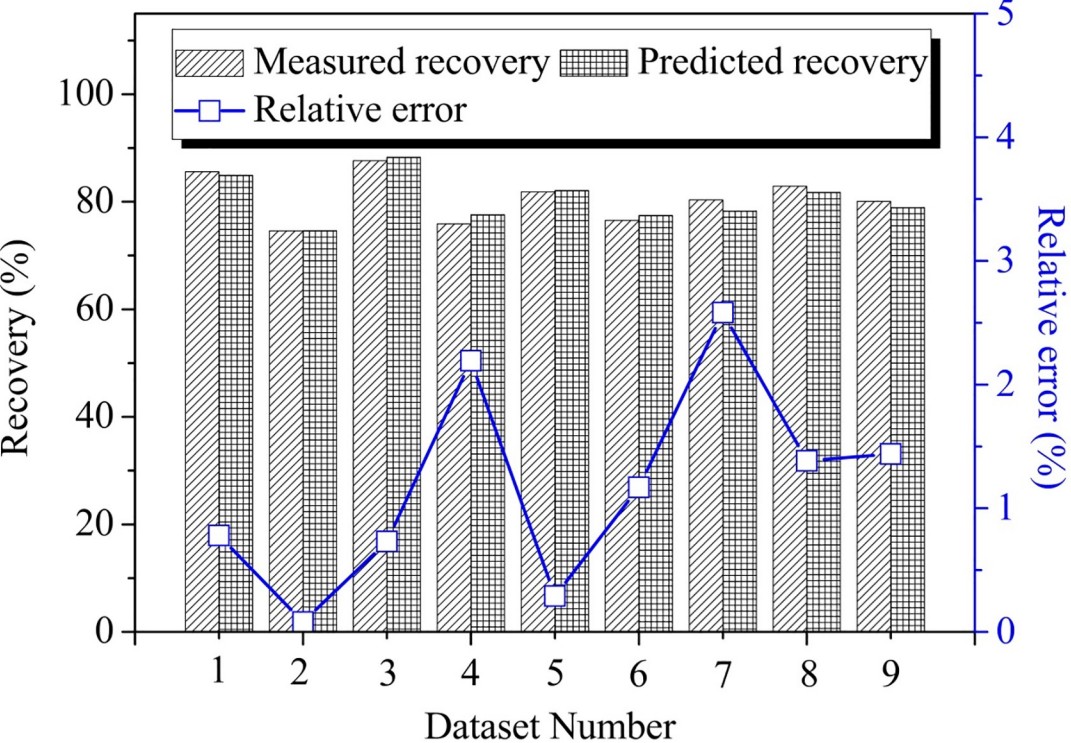

**Fig 8. Comparison between measured and predicted value.**

significant and the linear fitting method is appropriate for establishing a statistical model [35]. The results of the significance test for multivariate linear fitting showed that the significance test values for each independent variable were less than 0.1, which indicates significance among the variables; for each of the variables, all of the variance-inflation factors for each variable VIF were less than 5, which indicated no co-linearity between variables. The above analysis shows that, in a statistical sense, Eq (3) represents the relationship between the four dimensionless dimension groups and the recovery ratio very well.

9 groups of testing data were reserved from the experimental data, and 9 groups of forecast data were acquired using Eq (3). The results between a comparison of the 9 groups of forecast data and the 9 groups of experimental data are shown in Fig 8. In this figure, the predicted values were approximate to the actual ones, and the relative error curve (shown as a red line) had a maximum error of 2.58%. The above analysis suggests that Eq (3) Formula (3) can be effectively used to predict the recovery of different structural parameters in the process of mining of gently inclined ore bodies.

According to the physical simulation experiments, if the optimum industrial experimental parameters were determined as: stope length L = 30 m, production drift interval d = 6 m, ring burden 2 m, footwall side blasthole angle 40˚, and the footwall friction coefficient as the natural repose angle of the ore, the friction coefficient would be f = tan40 = 0.84. According to Eq (2), the ore-recovery ratio of the industrial experiment is expected to reach 81.97%.

## Case study

Jiaojia Gold Mine of Shandong Gold Group has a dip angle of 30˚and a vertical thickness of 12m. The mineralization of the ore body is uniform and stable with regular geometry, which

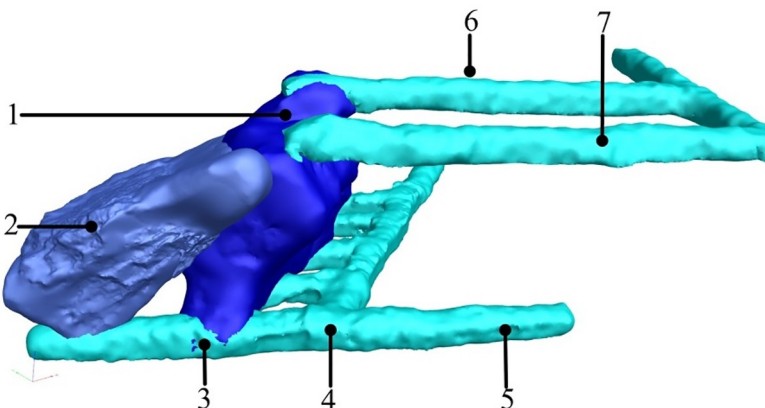

**Fig 9. CSM scanning results of field experiment.** 1 The void of lower stope after mining; 2 The void of upper stope after mining; 3 lower production drill drift; 4 sublevel transport drift; 5 stope ramp; 6 fill drift; 7 Mine prospection drift.

belongs to a typical inclined medium thick ore body. The quality of ore and rock can be classified as class I or II rock mass, and the stability of ore and rock is good. The upward horizontal cut and fill mining method that has long been used in this gold mine has many disadvantages, such as small production capacity, low production efficiency, more operation stopes, intensive manpower, and big losses at the triangle ore body of the hanging wall. The sublevel open stopping using long-hole with back filling mining method is designed in this mine experiment. The mining parameters are determined according to the structural parameters from the above dimensional analysis: the length of the stope is L = 30m; the height of the section is 10m; the width of the stope, which equals to the thickness of the ore body, is 12M; the drift interval is d = 6m; the ore drawing ring burden is 2m; and the side hole angle of the footwall caving is 40 degrees.

After mining, CMS 3D laser scanners were set up in the sublevel filling drift to conduct mining effect assessments, the results of which are shown in Fig 9.

As shown in Fig 10A, the losses in the lower chamber are due to three factors: (1) drift end residue, (2) hanging wall excavation loss, and (3) footwall residue losses. These parts account for a total loss ratio of about 18%, which means that the recovery ratio remains around 82%. The waste rock caving from the footwall due to the flow angle expansion operation results in a dilution ratio of 18.4%. As the grade of the footwall waste-rock was around 0.5 g/t, mixed waste-rock had little effect on ore dilution. In the extraction process in the upper subframe stope, the rock surrounding the hanging wall was subject to only sporadic caving and was quite stable on the whole. According to Fig 10B, it was calculated that the ore-loss ratio was 16.84%, and that the recovery ratio was 83.16%. The mixed waste-rock covered two main parts: scattered waste rock that has collapsed during the excavation and backfilling body from the lower sublevel stope, with a waste rock mix ratio of 6.3%. It should be noted that the field industrial experiment results were approximate, and the results were acquired from physical simulation and multiple linear fitting. Nevertheless, the results indicate that simulation experiments are useful for identifying the loss mechanism of sublevel open stoping using subframe long hole with back fill mining method.

## Conclusions and recommendation

### Conclusions

After ore drawing, the ore remnants in the lower sub frame stope mainly consisted of sidewall remnants at the footwall, ridge remnants between the access drifts, and end triangle remnants.

The sidewall remnants at the footwall were primarily influenced by the footwall side blast hole angle. As the footwall side blast hole angle increased, the amount of footwall remnant decreased; and the ore recovery ratio increased at first and then decreased, reaching its maximum value when the footwall side blasthole angle was 40˚. Ore recovery was also influenced by the degree of smoothness of the footwall surface; analysis of the results for different friction coefficients in the blasted ore drawing experiment revealed that the ore-recovery effect increased in proportion to the smoothness of the footwall. The relationship between loading drift spacing and production ring burden and ore loss during subframe stope was less obvious, and can be specifically selected according to production safety and efficiency needs.

Given the results of the physical simulation experiments and production requirements, it can be concluded that the optimum structure parameter combination for the maximum ore recovery ratio is as follows: a footwall side blast hole angle of 40˚, production loading drift intervals of 6.5 m, and a production ring burden of 2 m. Physical simulation tests showed that the above conditions yield a recovery ratio of up to 80%.

The results of the physical simulation tests show that ore recovery has a linear correlation with various factors. By using the SPSS multiple linear-regression analysis tools for dimensional analysis, a statistical model can be constructed that describes the ore recovery ratio, the footwall side blast hole angle, the flatness of the footwall after blasting, the number of trackless equipment going through the loading drift, and the number of blasts conducted over the course of the sublevel open stoping method. A comparison between the statistical model's prediction values and the physical simulation's experimental values shows that the model closely matches the physical simulation results, which makes it suitable for predicting the ore-recovery ratio during sublevel open stoping of inclined medium-thick ore bodies. Furthermore, on this basis, it is estimated that the ore-recovery ratio is 81.97% under the optimum structural parameters.

The sublevel open stoping using subframe long hole with back filling mining method was applied in practical production conditions, and a 3D laser scanning system CMS was used in the stoped area after extraction in order to evaluate the mining process. The industrial experimental results showed that ore production capacity can reach up to 600 t/d, the recovery ratio and dilution ratio of the lower chamber were 82% and 8.4% respectively, while the recovery ratio and the dilution ratio of the upper chamber were 83.16% and 6.3% respectively. The experimental results matched the prediction values of the physical simulation and the statistical model.

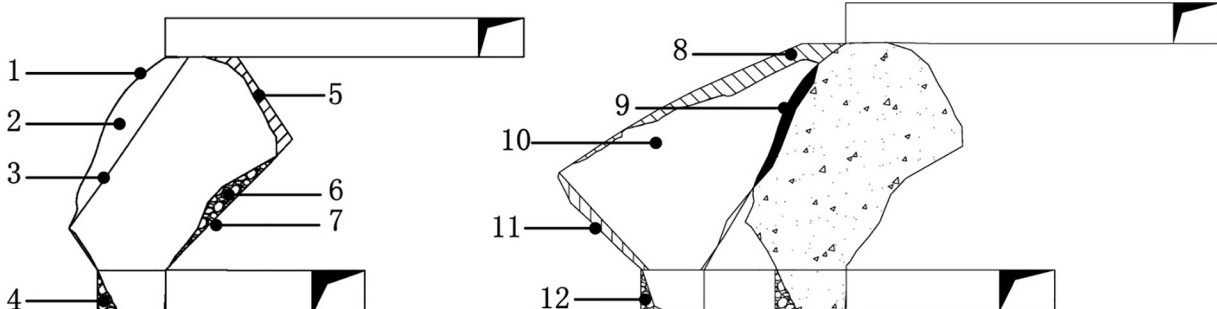

**Fig 10. Experimental stope loss value analysis results.** (A) Lower stope mining effect picture. (B) Upper stope mining effect picture. 1 Lower chamber mining boundary; 2 Lower chamber mining over-excavation part; 3. Designed upper boundary of the Lower chamber; 4 tunnel residue; 5 Lower chamber under excavation part; 6 Footwall remnant ore body; 7 Designed lower boundary of the Lower chamber; 8 Upper chamber under excavation part; 9 Over-excavation part of upper chamber at Lower chamber backfilling body; 10 Upper chamber exterior boundary; 11 Upper chamber under excavation part; 12 Upper chamber end residue.

## Recommendation

The sublevel open stoping using sub frame long hole with back filling mining method proposed in this paper is applicable to inclined medium thick ore body which is either stable as a whole ore body or with stable hanging wall. By using this method, if the footwall rocks contain part of the grade, the ore loss and dilution can be reduced in the mining process.

Optimizing design and controlling the drilling precision of blasting holes can ensure that a footwall surface that is as smooth as possible can be formed after the bursting of multiple positions. This is beneficial as it allows for a larger recovery ratio while using a relatively small footwall side blasthole angle. At the same time, it also ensures that a greater quantity of ore can be recovered with no additional engineering investment, which thus reduces the cost of mining.

Ridge remnants and end remnants are mainly influenced by production ring burden and LHD loading location, and can be eliminated by auxiliary mine extraction via the drilling drifts along the ore body, or with the aid of a remote-controlled LHD.

## Supporting information

**S1 File. Experiment data for all cases.**
(XLSX)

## Acknowledgments

The authors express sincere appreciation to the reviewers for their valuable comments and suggestions, which improved the paper much.

## Author Contributions

**Methodology:** Jin Wu.

**Writing – original draft:** Jin Wu.

**Writing – review & editing:** Jin Wu.

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
