## [Decision Letter · Decision Letter 0]

12 Mar 2020

PONE-D-20-00583

Research on Sublevel Open Stoping Recovery Processes of Inclined Medium-Thick Orebody on the Basis of Physical Simulation Experiments

PLOS ONE

Dear Mrs Wu,

Thank you for submitting your manuscript to PLOS ONE. After careful consideration, we feel that it has merit but does not fully meet PLOS ONE’s publication criteria as it currently stands. Therefore, we invite you to submit a revised version of the manuscript that addresses the points raised during the review process.

We would appreciate receiving your revised manuscript by Apr 26 2020 11:59PM. To enhance the reproducibility of your results, we recommend that if applicable you deposit your laboratory protocols in protocols.io, where a protocol can be assigned its own identifier (DOI) such that it can be cited independently in the future. For instructions see: http://journals.plos.org/plosone/s/submission-guidelines#loc-laboratory-protocols

We look forward to receiving your revised manuscript.

Kind regards,

Hongbing Ding, Ph.D.

Academic Editor

PLOS ONE

Additional Editor Comments (if provided):

Thank you for submitting your manuscript to PLOS ONE. The reviewers recommend reconsideration of your paper following minor revision. I invite you to resubmit your manuscript after addressing all reviewer comments.

Journal Requirements:

3. Please include a copy of Tables 1-5 which you refer to in your text on page 6, 7 and 10.

Reviewers' comments:

Reviewer's Responses to Questions

**Comments to the Author**

1. Is the manuscript technically sound, and do the data support the conclusions?

Reviewer #1: Yes

Reviewer #2: Yes

2. Has the statistical analysis been performed appropriately and rigorously? 

Reviewer #1: Yes

Reviewer #2: Yes

3. Have the authors made all data underlying the findings in their manuscript fully available?

Reviewer #1: Yes

Reviewer #2: Yes

4. Is the manuscript presented in an intelligible fashion and written in standard English?

Reviewer #1: Yes

Reviewer #2: Yes

5. Review Comments to the Author

Reviewer #1: The experiment appears to have been carefully designed and executed, and the results should be of interest to the readership of PLOS ONE. However, before I can recommend publication, the author should address some issues, as follows:

1. The lack of all tables might not contribute to my review. Please upload tables. I strongly suggest the new manuscript should be reviewed again.

2. Please discuss how relevant the boundary condition is to existing conditions in the field. For example, in the mines considered, how the characteristics of overlying material (harder, softer, more or less dense) impact the results? Or would the mixture of unwanted material in an actual mine affect to a significant extent the experimental results?

3. There is not any information about the mechanical characteristics of the material. How the strength of the rock impact the results?

4. The paper is well written but still need English improvements. I advise the author to find a native English speaker to proofread the manuscript. The readability of the paper could be greatly improved, and this would make it much more accessible to the readership of PLOS ONE.

Reviewer #2: (1) In chapter 2, the author should add some sketches to make "the sublevel open stoping using long-hole with back filling mining method" clearer.

(2) In chapter 3, the site conditions should be explained clearly in the similar simulation experiment;

(3) Please explain why the model is constructed with iron plate and acrylic plate and whether it will affect the simulation results.

6. PLOS authors have the option to publish the peer review history of their article (what does this mean?). If published, this will include your full peer review and any attached files.

Reviewer #1: No

Reviewer #2: No

---

## [Author Response · Author response to Decision Letter 0]

17 Apr 2020

Authors’ reply to the Editor’s and Reviewers’ questions/comments/suggestions:

The authors wish to greatly appreciate the invaluable and constructive comments, suggestions and contributions of the Editor and Reviewers to improve the quality of their research paper. All points highlighted by editor and reviewers have been fully addressed and clarified. Please find below the authors’ reply to the Editor’s and Reviewers’ comments/questions/suggestions:

(Authors’ reply: Blue in colour)

Reviewer #1:

1. Please include a copy of Tables 1-5 which you refer to in your text on page 6, 7 and 10.

Authors: Sorry for the mistake. Tables 1-5 have been added in the corresponding part of the text.

2. Please discuss how relevant the boundary condition is to existing conditions in the field. For example, in the mines considered, how the characteristics of overlying material (harder, softer, more or less dense) impact the results?

Authors: To guarantee that the results of a physical model can be directly scaled up, the model must be fully similar to the mine scale. Though physical model and full scale model can not meet the similarity in all aspects, geometric similarity is the most important and the easiest to implement in physical ore drawing simulation model.

(1) Conditions of similarity

To mimic the actual conditions in the field, the properties of materials used in the physical model were determined according to the following conditions of similarity:

a) Length scale CL

b) Time scale Ct

C) Stress scale

Through lab and in-site tests, the natural rest angle of the granules flow is consistent with the natural resting angle of the blasting ore, indicating that the mechanical conditions of the granules flow are consistent.

DLSOS mining method which is different from caving mining method (sublevel caving or block caving), belongs to open stope mining method. Its mining sequence is from bottom to top. Because the blast ore is mined under the protection of the top pillar, the blast muck is not affected by the volume weight of overlying strata, and the stress will not be transferred. Therefore, it is not necessary to apply the corresponding load on the upper part of the physical model.

(2) Determination of the parameters employed in physical model

The testing ore material for the physical simulation experiment is collected from the field. Therefore, the density （Cρ=1）and hardness of the ore material of the model are the same as those in the field. During the experiments, the test ore material is collected with the following procedure.

the ore granularity is examined;

the ore taken from the field will be crushed and screened according to the granularity listed in Table 1. And the ore will be collected by different particle sizes. 

given the set geometric similarity coefficient k, the gross mine weight G for simulation will be calculated. With the given abundant factor as 1.2, the ore volume with different granularity will be determined. The required weight of each particle size can be calculated according to the weight proportion of each particle size shown in Table 1. After weighing and mixing, the ore and waste rock meeting the requirements of particle size combination are prepared. 

In the experiment, the artificial removal of ore was used to simulate the field ore transportation. The speed of one-time manual transportation is calculated according to the following process. According to the time similarity coefficient, the time to complete one ore drawing is determined.

The time of drawing operation satisfies the similarity coefficient: k1/2.

In mining experiment, the ore quantity of one blasting is 750 t, the efficiency of 3 m3 LHD is 500 t/shift, and the time of ore extraction after blasting is 1.5 shift. The mining time after blasting is 1.5 shifts. If each shift lasts 8 hours, the ore drawing time is 12 hours at a time.

The ore yield per hour is 62.5 t. According to the time similarity ratio, the length similarity ratio is k = 50, and Ct = 501/2 = 7.07. Then the time of the physical ore drawing experiment when one burden is moved is calculated as below:

the time is 12/ Ct =12/7.07 = 1.697 h.

In the ore flow experiment, the loading capacity of the experimental model is 36.05 kg, with a total of 14 burdens. One burden loading capacity is 2.58 kg, and the drawing time is 1.697 hours. The ore drawing speed is required to be 25.34 g/min.

Therefore, the length, width and depth of the small shovel for drawing are 3 cm, 2 cm and 1 cm respectively. Ore draw quantity is 9g/circle and frequency is 2.8 times per minute. Based on the drawing frequency, the physical simulation drawing time can be guaranteed to be consistent with the in-site drawing time.

The physical simulation test ensures the consistency of time similarity and in-site experiment by controlling it as 2.58 times/min.

3. Or would the mixture of unwanted material in an actual mine affect to a significant extent the experimental results?

Authors: The mixing of waste rock has a great influence on the real industrial experiment results. In the actual mining production process, we do not want to mix waste rock into the ore, although the mixing of waste rock is inevitable. Because the mining method proposed in this paper is the open stope mining method, the prerequisite for the implementation of this mining method is that the stope is stable and the footwall and hang wall are stable. Therefore, the waste rock will not fall and mix into the ore falling in the room, thus reducing the quality of the ore. In the industrial experiment, the surrounding rock mass of the hanging and foot wall belongs to class I and II rock mass, which is of good quality. Under the selected structural parameters, the stope can remain stable without the rock falling from the hanging and foot wall and mixing into the ore.

In this paper, on the one hand, based on the ideal conditions of this mining method, waste rock does not mix in the ore. On the other hand, it is difficult to simulate the time and location of waste rock mixing in laboratory experiments. Therefore, in the selection of experimental indicators, the mixing rate of waste rock is not considered. And only the ore recovery rate is considered to study the location and quantity of mine loss, which provides support for the selection of parameters in industrial experiments.

This information is summarized on page 6, lines 187-202.

4. There is not any information about the mechanical characteristics of the material. How the strength of the rock impact the results?

Authors: This is a very good question. This problem involves a precondition of physical simulation experiment of bulk flow. The most important factor affecting ore flow in ore drawing experiment is particle size composition. The fluidity of ore particles with different sizes is different. The ore fluidity will directly affect the recovery quantity of ore, leading to the difference of the location and quantity of ore loss in the mining process. Thus it will cause the difference of mining efficiency and mining cost.

In the real mining, ore particles will collide with each other in the flow process. During the collision, the ore with low strength will be further broken, leading to the change of particle size distribution, and then affecting the flow behaviour of ore. However, this process usually occurs in the mining method or ore drawing pass with large ore rock movement distance. In the application of the mining method adopted in this paper, the movement of ore and rock is continuous and slowly downward. In the real production process, the possibility of large distance and high-speed collision of ore and rock is very small. In terms of the mine considered, the uniaxial compressive strength of the ore and rock is greater than 70MPa, and the uniaxial compressive strength of the rock is more than 40MPa. In this case, the rock is relatively strong, and there is little mutual expansion and fragmentation in actual production. Furthermore, the ore material used in the lab experiment is obtained directly from the site, thus the mechanical properties of the ore are the same as the real ore on the site. The particle grading is formulated according to the similarity ratio. The particles are relatively hard, and the collision will not lead to rock fragmentation. 

Therefore, for the lab experiment, the impact of the mechanical properties of the ore on the recovery can be ignored.

5. The paper is well written but still need English improvements. I advise the author to find a native English speaker to proofread the manuscript. The readability of the paper could be greatly improved, and this would make it much more accessible to the readership of PLOS ONE.

Authors: As suggested, we have carefully checked the language of the manuscript including grammar and expression. With the help of a native English speaker in this major, the language has been polished.

Thank you for your constructive and valuable review comments.

Reviewer #2:

1. In chapter 2, the author should add some sketches to make "the sublevel open stoping using long-hole with back filling mining method" clearer.

Authors: Thank you for your suggestions. The original illustration of mining method (Figure 2) is a two-dimensional plan, which is not clear and intuitive enough when describing mining methods. As suggested, Figure 2 has been updated: the former planar graph remains as Figure 2 (b); and an additional three-dimensional map of mining method is added as Figure 2 (a). Fig. 2 (a) and Fig. 2 (b) are related by section lines I-I and Ⅱ-Ⅱ, which should be easier for readers to understand. This information is given on page 4 and 5, lines 106-144 and page 26 Fig.2(a)

2. In chapter 3, the site conditions should be explained clearly in the similar simulation experiment.

Authors: Thank you for this good suggestion. The following information regarding the conditions of the site is provided on page 12 and 13, lines 393-396.

Jiaojia Gold Mine of Shandong Gold Group has a dip angle of 30°and a vertical thickness of 12m. The mineralization of the ore body is uniform and stable with regular geometry, which belongs to a typical inclined medium thick ore body. The quality of ore and rock can be classified as class I or II rock mass, and the stability of ore and rock is good. The sublevel open stopping using long-hole with back filling mining method is designed in this mine experiment. The mining parameters are determined according to the structural parameters from the above dimensional analysis: the length of the stope is L = 30m; the height of the section is 10m; the width of the stope, which equals to the thickness of the ore body, is 12M; the drift interval is d = 6m; the ore drawing ring burden is 2m; and the side hole angle of the footwall caving is 40 degrees. The parameters are consistent with the prototype parameters studied in the physical ore drawing simulation experiment, and the ore rock used in the experiment is directly taken from the mine experimental room; its particle size composition is also obtained based on the analysis of the real particle size of the mine. Through the above process, the similarity between the physical ore drawing simulation experiment and the real industrial experiment is ensured.

3. Please explain why the model is constructed with iron plate and acrylic plate and whether it will affect the simulation results.

Authors: (1) In the physical simulation experiment of bulk flow, the acrylic plate which is a white transparent material, is used as the external boundary of the model. During the experiment, the flow process of ore particles and the residual situation of ore can be observed from the outside of the model. Secondly, the acrylic plate characterized by high hardness, strength and toughness, is not easy to deform and suitable to process. Thus, the acrylic plate is employed in the construction of the ore drawing model. 

The iron plate is used as a division plate of ring burden to simulate different blasting operations and to control the amount of ore caving. In the process of ore drawing experiment, the ring burden plate needs to be removed, so the rigidity and toughness of iron plate are suitable for drawing. Thus, iron plate is selected as the isolation plate in ore drawing experiment.

(2) In the industrial experiment, the rock surface of footwall formed by two adjacent blastholes is usually uneven. In the process of lab ore drawing physical experiment, in order to study the influence of the flatness of the footwall surface on the flow of the caving ore after blasting, the flatness of the footwall rock surface in actual production is simulated by placing the pasted fine sandpaper, coarse sandpaper and ore particles on the inner surface of the acrylic plate of the footwall (the side contacting the ore). The friction coefficient corresponding to different flatness is usually determined by the sliding experiment of inclined surface in classical mechanics. The method is shown in the figure below. Therefore, the acrylic material itself will not affect the ore fluidity.

In mining production, the blasting is carried out by upward medium deep holes. After blasting in medium deep holes, the blasting working faces (end position) of adjacent two rows are usually relatively flat. And when iron plate is used to simulate the division plate, the corresponding end position is the end position. Therefore, the influence of iron plate as the isolation plate on the ore fluidity can be ignored.

Thank you for your constructive and valuable review comments.

---

## [Editor Report · Decision Letter 1]

20 Apr 2020

Research on Sublevel Open Stoping Recovery Processes of Inclined Medium-Thick Orebody on the Basis of Physical Simulation Experiments

PONE-D-20-00583R1

Dear Dr. Wu,

We are pleased to inform you that your manuscript has been judged scientifically suitable for publication and will be formally accepted for publication once it complies with all outstanding technical requirements.

With kind regards,

Hongbing Ding, Ph.D.

Academic Editor

PLOS ONE

Additional Editor Comments (optional):

The authors have done a good job in revising the manuscript. Now it can be accepted for publication in PLOS ONE.
---

## [Editor Report · Acceptance letter]

24 Apr 2020

PONE-D-20-00583R1 

Research on Sublevel Open Stoping Recovery Processes of Inclined Medium-Thick Orebody on the Basis of Physical Simulation Experiments 

Dear Dr. Wu:

I am pleased to inform you that your manuscript has been deemed suitable for publication in PLOS ONE. Congratulations! Your manuscript is now with our production department. 

With kind regards,

on behalf of

Professor Hongbing Ding 

Academic Editor

PLOS ONE